# Targeted Mitochondrial Epigenetics: A New Direction in Alzheimer’s Disease Treatment

**DOI:** 10.3390/ijms23179703

**Published:** 2022-08-26

**Authors:** Ying Song, Xin-Yi Zhu, Xiao-Min Zhang, He Xiong

**Affiliations:** Department of Pharmacology, Zhejiang University of Technology, Hangzhou 310014, China

**Keywords:** mitochondrial epigenetics, Alzheimer’s disease, mitochondrial DNA, methylation, noncoding RNA, post-translational modification

## Abstract

Mitochondrial epigenetic alterations are closely related to Alzheimer’s disease (AD), which is described in this review. Reports of the alteration of mitochondrial DNA (mtDNA) methylation in AD demonstrate that the disruption of the dynamic balance of mtDNA methylation and demethylation leads to damage to the mitochondrial electron transport chain and the obstruction of mitochondrial biogenesis, which is the most studied mitochondrial epigenetic change. Mitochondrial noncoding RNA modifications and the post-translational modification of mitochondrial nucleoproteins have been observed in neurodegenerative diseases and related diseases that increase the risk of AD. Although there are still relatively few mitochondrial noncoding RNA modifications and mitochondrial nuclear protein post-translational modifications reported in AD, we have reason to believe that these mitochondrial epigenetic modifications also play an important role in the AD process. This review provides a new research direction for the AD mechanism, starting from mitochondrial epigenetics. Further, this review summarizes therapeutic approaches to targeted mitochondrial epigenetics, which is the first systematic summary of therapeutic approaches in the field, including folic acid supplementation, mitochondrial-targeting antioxidants, and targeted ubiquitin-specific proteases, providing a reference for therapeutic targets for AD.

## 1. Introduction

Dementia affects more than 55 million people worldwide, with an average of one new case of dementia every three seconds [1]. Alzheimer’s disease (AD) accounts for 50–60% of the total cases of dementia patients. AD is a chronic neurodegenerative disease characterized by progressive cognitive impairment and memory loss, accompanied by different degrees of psychobehavioral symptoms, such as insomnia, depression, and irritability [2]. AD has become a public health problem that cannot be ignored, bringing enormous mental and economic pressures not only to families but also to the world economy [3].

Numerous studies have shown that epigenetics [4,5,6,7] plays an important role in AD, but the study of mitochondrial epigenetics is relatively superficial. We focus on and summarize recent research on mitochondrial epigenetics in AD, hoping to further elucidate the pathogenesis of AD and find new targets for the treatment of AD.

## 2. Pathological Features of AD

AD is a complex neurodegenerative disease whose pathogenesis is still not fully understood, and its main pathological features include the formation of age spots, β amyloid deposition in the brain, Neurofibrillary tangles (NFTs) due to the hyperphosphorylation of the tau protein, and neuronal death [8]. The clinical manifestations of patients with AD mainly include the following aspects: (1) progressive memory impairment or amnesia; (2) spatial disorientation; (3) language disorders dominated by fluent aphasia; (4) aberration and misuse; (5) intellectual disability; (6) personality changes; and (7) psychiatric symptoms [9]. As research progressed, more than a dozen diseases, including type 2 diabetes [10,11], hypertension [12,13], obesity [14], Down syndrome [15], hearing impairment [16], vitamin deficiency [17,18], and other diseases (Figure 1) increase the risk of AD.

The proposed AD pathogenic hypothesis includes the “amyloid cascade hypothesis” [19], the “cholinergic hypothesis” [20], neuroinflammation [21,22], metabolic disorders [23,24], and vascular injury [25]. Based on these, scientists hope to prevent and treat Alzheimer’s disease by inhibiting Aβ production, fibrosis, deposition, and inhibiting cholinesterase activity [26,27], but the effects of such treatments are not satisfactory. Existing drugs can only delay the aggravation of pathological symptoms, but cannot stop or reverse the disease process, which commonly recurs after stopping the drug. This cannot help but make one wonder whether current cognition is really part of the root cause of Alzheimer’s disease, or whether it is hindered by its appearance and ultimately cures the symptoms but not the root causes.

Recently, a recent study from the Grossman School of Medicine at New York University in the United States and the Nathan Klein Institute [28] subverted the traditional impression of Alzheimer’s disease that “extracellular amyloid plaques are formed first, followed by nerve cell death”, proposing that “nerve cell death first, and then extracellular amyloid plaque appear”. This view has opened up new ideas and research directions for the current dilemma that the efficacy of AD drugs is not obvious.

It is well-known that mitochondria are the “energy factories” of cells. Mitochondrial dysfunction is one of the non-negligible causes of nerve cell death. Altered gene expression and ATP production disorders due to mitochondrial epigenetic changes can lead to a variety of disorders [29], including age-related neurodegenerative diseases [30,31,32], metabolic alterations [33], altered circadian rhythms [34], and cancer [35,36]. Therefore, understanding the role of mitochondrial epigenetics in AD has important theoretical and clinical significance for analyzing the pathogenesis of AD.

## 3. Mitochondrial Epigenetic Changes in AD

### 3.1. Mitochondrial Genome and Mitochondrial Epigenetics

Mitochondria are semiautonomous organelles with unique DNA and genetic codes, which not only play an important role in amino acid synthesis, lipid metabolism, energy generation, and other metabolic processes, but also in free radical production, calcium homeostasis, cell cycle regulation, apoptosis, and signal transduction life homeostasis [37].

Human mitochondrial DNA (mtDNA) is a double-stranded closed-loop structure 16,569 bp in length (Figure 2), with both strands having independent coding and noncoding regions [38]. The mitochondrial coding region genome contains 37 genes, 13 of which encode polypeptides required by the electron transport chain (ETC), including 7 genes encoding complex I subunits (*MT-ND1, MT-ND2, MT-ND3, MT-ND4, MT-ND4L, MT-ND5*, and *MT-ND6*), a gene encoding complex III (*MT-CYTB*), 3 genes encoding complex IV (*MT-COI*, *MT-CO II*, and *MT-CO III*) and 2 encoding complex V (*MT-ATP6* and *MT-ATP8*), as well as two encoding ribosomal RNAs (rRNAs) (*12S rRNA* and *16S rRNA*) and 22 encoding transport RNAs (tRNAs) [39]. The *Displacement loop (D-loop)* is a unique noncoding region of mtDNA that controls the replication and expression of the mitochondrial genome [40]. The mitochondrial genome coregulates the bioenergy of cells with the nuclear genome. Thus, mitochondrial genomic variants can directly or indirectly influence all energy-dependent cellular processes and shape the metabolic state of the organism.

The genome consists of two types of genetic information, genetic information and epigenetic information provided by DNA sequences. Epigenetics refers to the association of epigenetic modifications with the regulation of gene expression and differentiation, as well as genetic changes in gene activity or cell phenotypes, without altering the DNA sequence [41]. However, compared to nuclear genes, mtDNA lacks the protection of histones and complex DNA repair mechanisms [42] and is susceptible to oxidative stress, resulting in changes in mtDNA mutation, mitochondrial epigenetic, and copy number. Reactive oxygen species (ROS) are the determinants of DNA methylation alterations [43] (Figure 3). Mitochondrial epigenetic modifications include DNA methylation, noncoding RNA, and post-translational modifications of nucleoid-related proteins [44].

### 3.2. mtDNA Methylation in AD

DNA methylation refers to the process of adding methyl groups from S-adenosine-methionine to a DNA base under the catalysis of DNA methyltransferase (DNMT). DNMT1 and DNMT3A are present in both the nucleus and mitochondria, while DNMT3B is found only in the nucleus [45]. DNMT1, DNMT3A, and DNMT3B are responsible for base methylation, and when cytosine methylation occurs at the C-5 position, it produces 5-methylcytosine (5 mC), the methylation of the NH2 group at the outer ring C-6 position of the adenine, and conversion to N6-methyladenosine (N6-methyladenosine, 6 mA) [46]. The transcription factors NRF1 and PGC1-α, activated by oxidative stress, have been shown to increase the expression of DNMT1, perhaps modulated by changes in methylation caused by oxidative stress [47,48].

mtDNA methylation has been found in a variety of environmental factors and endogenous metabolites [49], such as environmental exposures [50,51], diabetic retinopathy [33,52], cardiovascular disease [53,54], colon cancer [55,56], cerebral autosomal dominant arteriopathy with subcortical infarcts and leukoencephalopathy [57], and neurodegenerative diseases [30,32]. Although there is no direct evidence that mtDNA methylation is associated with the onset and development of these diseases, mtDNA methylation may be critical in many pathophysiological processes by regulating mitochondrial gene expression. We already know that Aβ42 acts as an AD toxic protein that destroys mitochondrial membrane permeability and affects the normal function of the mitochondria, leading to cell death [58]. In this process, there are also changes in mtDNA methylation (Figure 4). Therefore, mtDNA methylation is most likely a potential mechanism leading to AD mitochondrial dysfunction.

Summarizing the recent studies on the methylation changes of mtDNA in AD (Table 1), we found that the methylation of the *D-loop* region, *12S rRNA*, *CYTB*, *COX II*, *MT-ND1*, and other coding genes of mtDNA are altered in AD. In particular, there have been many reports of methylation changes in the D-loop region. Researchers have observed that, in the hippocampal tissue of APP/PS1 transgenic mice, in peripheral blood samples from LOAD patients, patients with early AD and advanced AD, the D-loop region was demethylated, the *12S rRNA*, *CYTB*, and *COX II*-encoding genes were hypermethylated, and the mtDNA copy number was reduced. However, Blanch et al. observed the opposite phenomenon, with the hypermethylation of the *D-loop* region and low methylation of *MT-ND1* in the endosteal cortex and the cerebral cortex of APP/PS1 mice in brain samples of Alzheimer’s disease-associated pathologies.

The mitochondrial D-loop region is the only noncoding region in mtDNA, measuring 1124 bp (positions 16024-576) and containing the necessary replication and transcription elements [65], also known as the mtDNA control region. The D-loop region is also the most susceptible area on mtDNA to oxidative damage [66]. The mitochondrial 12S rRNA gene encodes a protein that promotes secondary RNA formation, mitochondrial ribosome assembly, and mitochondrial translation, and plays an important role in the normal functioning of the mitochondria, and methylation of the *12S rRNA* gene reduces the expression of 12S rRNA, resulting in mitochondrial ribosome failure and a decrease in mtDNA copy number [67]. The *MT-CYTB* and *MT-COX II* genes are important coding genes for mitochondrial respiratory chain complex III and complex IV, respectively. In general, DNA hypermethylation inhibits gene expression, whereas DNA demethylation induces gene reactivation and expression [68,69]. Based on the available results, we believe that in AD, mtDNA hypermethylation causes damage to mitochondrial biogenesis and the normal functioning of the mitochondrial respiratory chain, and that the demethylation of the D-loop region is likely a compensatory mechanism for this damage [70,71]. Interestingly, AD specimens from different analytical tissues or different stages of the disease may show different results. Better generalizing these differences will provide accurate biomarkers for AD.

However, there are still many difficulties in mtDNA detection. For example, the existence of an mtDNA secondary structure and mitochondrial pseudogenes interferes with the detection accuracy of mtDNA methylation, copy number, and biogenesis. The studies presented in Table 1 were sequenced using bisulfite pyrosequencing, 454 GS FLX Titanium pyrosequencing, and methylation-sensitive high-resolution melting (MS-HRM) methods, respectively. The 454 GS FLX Titanium pyrosequencing process assesses methylation at CpG and non-CpG sites [60]. The number of unmethylated reads was higher than the number of methylated reads per identified site, and few methylated sites were lost. Therefore, the researchers removed from the analysis those reads that showed at least one missing site in the methylation pattern after alignment to avoid bias in quantification. This method effectively avoids the analysis of putative mitochondrial pseudogenes and ensures that the amplicons are 100% identical to mtDNA. bisulfite pyrosequencing and MS-HRM also need to eliminate the influence of mitochondrial pseudogenes by certain means, such as isolating mitochondria before mtDNA extraction to avoid nuclear contamination, and specific primers considering NUMT amplification Basic Local Alignment Search Tool searches to identify known NUMTs [72].

### 3.3. Mitochondrial Noncoding RNA Modification in AD

RNA modification is another way of regulating gene expression. Noncoding RNAs (ncRNAs) are RNAs [73] that are transcribed from the genome, not translated into proteins, and can perform their respective biological functions at the RNA level [73], most commonly tRNA and rRNA [29].

More than a decade ago, researchers found a lack of taurine modification in mutant mt-tRNAs in patients with mitochondrial disease, the first evidence of molecular pathogenesis caused by disorders of mitochondrial RNA modification [74]. In addition to taurine modifications, recently, Andrew M. Shafik’s team mapped brain-rich N1-methyladenosine (m1A) RNA modifications in the 5XFAD cortex of a mouse model of Alzheimer’s disease using the newly developed m1A-quant-seq method, finding that m1A levels in mitochondrial and cytoplasmic tRNAs are modulated in Alzheimer’s disease. Loss of m1A methyltransferase leads to a more harmful AD phenotype [75]. Variable hypermethylation affecting the translation of downstream mitochondrial proteins was observed at the ninth position (p9) of mt-tRNA in cerebellar tissue after death in elderly AD patients, and a significant correlation between nuclear gene expression and p9 methylation was found [76]. mt-tRNA methylation may affect the expression of nuclear genes through retrograde signaling, but the mechanism behind this is unclear [77,78].

### 3.4. Post-Translational Modification of Mitochondrial Nucleoid-Associated Protein in AD

Although mtDNA is not protected by histones, it is not bare [79]. mtDNA can form protein–DNA complexes with proteins. The structure of this complex is similar to the nucleus, so it is called a nucleoid, and the proteins in it are called nucleoid-associated proteins. The initiation mechanism of mitochondrial transcription consists of three nucleoid-associated proteins: RNA polymerase (POLRMT) and two cofactors, transcription factors A and B2 (TFAM and TFB2M) [80]. Nucleoid-associated proteins alter activity through post-translational modifications (PTM), which affect the normal function of mitochondria.

TFAM is a protein encoded by the nucleus of a cell and is an mtDNA-binding protein that plays an important role in maintaining the stability of the mitochondrial genome [81]. Several studies have reported that TFAM can induce phosphorylation [82,83] glycosylation [84], ubiquitination [85], and other PTM. Sean D. Reardon [86] tested the effects of TFAM phosphorylation by protein kinase A phosphorylation and TFAM by acetyl-CoA acetylation on DNA packaging, transcription initiation, and synthetic capacity in mitochondria, and found that phosphorylation and acetylation modifications did not affect TFAM’s ability to initiate the replication and compaction of mtDNA, but were able to enhance transcriptional synthesis. This finding is consistent with the views of Grace A. King [87]. In PD in vitro models, the extracellular regulated protein kinase (ERK) phosphorylation of another site of TFAM impairs TFAM’s ability to bind to mtDNA light-chain promoters [88]. In diabetes, researchers have found that TFAM is ubiquitinated to reduce its transport to mitochondria [89]. The decreased expression of mitochondrial biogenesis-related proteins PGC-1α and TFAM was also observed in AD cell models [90,91]. Diabetes is a major risk factor for triggering AD, and while its mechanism is unclear and direct evidence is lacking, we have reason to believe that TFAM is present in AD.

In addition to TFAM, TFB2M also shows phosphorylation. TFB2M is a protein necessary for promoter melting at transcription initiation [92] and has three phosphorylation sites, namely, threonine 184, serine 197, and threonine 313. Alicia M. Bostwick [93] found that the phosphorylation of these sites impairs promoter binding and prevents transcription.

## 4. AD Treatments by Mitochondrial Epigenetics

### 4.1. Folate Supplements

Folate is the most important methyl contributor in the human body, being able to transfer methyl groups to epigenetic DNA and enzymes and playing a key role in maintaining the normal function of DNA and mtDNA [94]. Folate deficiency affects the methylation levels of genes associated with oxidative stress, which in turn damages DNA and mtDNA, leading to cognitive impairment and even AD [95]. Folate deficiency also leads to the hypomethylation of promoter regions such as the D-loop region, which is one of the important mechanisms behind AD.

Furthermore, folate also modifies mitochondrial tRNAs. After Raphael J. Morsche [96] knocked out mitochondrial folate serine hydroxymethyltransferase-2, the lack of methylation modification of mitochondrial tRNAs in human cells led to translation defects and impaired the expression of respiratory streptose. This suggests that folate methylates mitochondrial tRNA by binding to a single carbon unit, a modification necessary for mitochondrial translation and oxidative phosphorylation.

Folate supplements have been shown to improve cognitive impairment in patients in the short term [97].

### 4.2. Mitochondria-Targeted Antioxidants

As already explained, oxidation stress is the most important mechanism for mitochondrial epigenetic changes, so the use of exogenous antioxidants is considered a promising strategy for blocking ROS-induced damage and countering the adverse mitochondrial epigenetic changes caused by AD. Traditional antioxidants such as vitamin C [98,99] and vitamin E (vit E) [100] have been used to study whether they have therapeutic effects on AD, but the results are not satisfactory. This is mainly because traditional antioxidants cannot localize to the mitochondria and cannot improve the damage to the mitochondria from oxidative stress at the root.

To improve the mitochondrial localization of antioxidants, mitochondrial-targeted antioxidants such as mitoquinone (mito Q) [101,102], mito Vit E [103], and mito TEMPO [104,105] have been developed. Mito Q, mito Vit E, and mito TEMPO consist of ubiquinone, vit E, antioxidant piperidine nitrate TEMPO, and decayed carbon chains of lipophilic triphenylphosphine (TPP) cations, respectively [106,107]. TPP promotes the entry of ubiquinone, Vit E, and TEMPO into the mitochondrial matrix and accumulates in the mitochondria to achieve antioxidant effects that target mitochondria. Although these mitochondria-targeting antioxidants are not effective in clinical treatment in patients with AD, this may be a favorable reference for future use in combination therapies with AD.

### 4.3. Targeted Ubiquitin-Specific Proteases

Ubiquitin is a highly conserved protein modifier that is ubiquitous in eukaryotic cells [108]. Post-translational modifications of ubiquitinated proteins have been found in pathogenic proteins in patients with AD, including mitochondrial proteins [109]. An overdose of ubiquitin-specific protease (USP) 25 is a key factor in inducing AD brain microglia homeostasis caused by Down syndrome [110]. Further studies demonstrated that USP25 ubiquitinated amyloid precursor protein and beta-site APP-cleaving enzyme 1 in Golgi apparatus lead to an amyloid pathology, while USP25 pharmacological inhibition can reverse amyloid pathology. Unlike USP25, which localizes the Golgi apparatus and promotes the ubiquitination of AD toxic proteins, USP30 is a deubiquitinating enzyme that is localized in the outer membrane of the mitochondrial body [111]. The deubiquitinating activity of USP30 is necessary to maintain the normal morphology of mitochondria and has important implications for mitochondrial dynamics [112]. The use of specific USP30 inhibitors can promote the autophagy of mitochondria [113]. USP is a new potential target for AD therapy.

## 5. Conclusions and Future Perspectives

Different mitochondrial epigenetic mechanisms play crucial roles in the occurrence and development of AD, and the disruption of the dynamic balance of mtDNA methylation and demethylation leads to damage to the mitochondrial electron transport chain and the obstruction of mitochondrial biogenesis, which is the most studied mitochondrial epigenetic change. The D-loop region, the regulatory center for mtDNA replication and transcription, was found to be demethylated in most AD animal models or human samples, while the *12S rRNA*, *CYTB*, and *COX II*-encoded genes were hypermethylated with decreased mtDNA copy numbers. *D-loop* demethylation may compensate for the hypermethylation of the *12S rRNA*, *CYTB*, and *COX II* coding genes. Still, researchers have not yet figured out whether the mtDNA methylation changes are the result of AD or the cause of the AD process. Further research is required to elucidate this.

In addition, mitochondrial ncRNA modifications and the PTM of mitochondrial nucleoid-associated proteins have also been observed in neurodegenerative diseases and related diseases that increase the risk of AD. There are still relatively few mitochondrial ncRNA modifications and mitochondrial nucleoid-associated proteins PTMs reported in AD, and we have reason to believe that this is an important factor affecting the development of AD, through mtDNA-specific transcription factor PTMs such as TFAM and TFB2M, which reduce protein expression. This review provides a new research direction for the AD mechanism, starting from mitochondrial epigenetics.

However, there are still some difficulties that cannot be ignored in mitochondrial epigenetic studies, such as false-positive and false-negative results of mtDNA methylation caused by the mtDNA secondary structure and mitochondrial pseudogenes [114]. How to be sure that what is being detected is true mtDNA, rather than mitochondrial pseudogenes, is a challenge researchers must overcome. Numts contamination is an important reason for the controversial application of bisulfite-pyrosequencing to mtDNA [115]. The detection of mtDNA methylation by bisulfite-pyrosequencing requires certain means to eliminate sequences with uncertain reads due to numts. Next-generation sequencing technology, such as 454 GS FLX titanium pyrosequencing, can effectively eliminate the influence of mitochondrial pseudogenes on the detection results, but this requires better equipment. Therefore, the development of more convenient, inexpensive, and efficient mtDNA methylation detection methods is the key to promoting the progress of mitochondrial epigenetics in AD.

This review summarizes therapeutic approaches that target mitochondrial epigenetics, which is the first systematic summary of treatments in this field, including folate supplements, mitochondrial-targeted antioxidants, and targeted USPs (Figure 5). As the most important methyl donor in the body, folate can not only transfer methyl groups to DNA but also transfer methyl groups to RNA to maintain the normal functioning of the mitochondrial genome. Antioxidants that target mitochondria can reduce ROS damage to mitochondria and reduce mitochondrial epigenetic changes at the source. The inhibition or maintenance of USPs can mediate the activity of mitochondrial proteins or AD toxic proteins, which has achieved the purpose of reversing the AD process.

However, the above treatment routes have shortcomings: (1) The duration of folic acid supplementation treatment is short, the optimal folic acid dose is unclear, and folic acid may need to be synergistic with other AD treatment drugs [116]; (2) although USP30 inhibitors have been shown to alleviate mitophagy impairment in Parkinson’s syndrome [117], it remains to be demonstrated whether USP30 inhibitors are effective in AD.

In conclusion, our work provides possible biomarkers for the clinical detection of AD and clarifies the mechanism of action of mitochondrial epigenetic modifications in AD, in an attempt to find AD treatment options that target mitochondrial epigenetics (Figure 4). In addition, we think more in-depth research on mitochondrial ncRNA modifications and the PTMs of mitochondrial nuclear proteins is required to yield unexpected gains. Both the clinical treatment of AD and the scientific research of this review are of great significance.

## Figures and Tables

**Figure 1 ijms-23-09703-f001:**
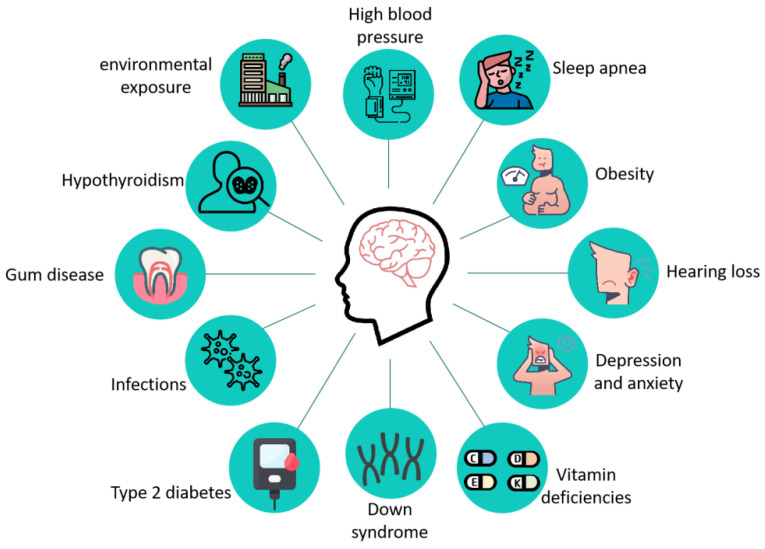
Related diseases that increase the risk of developing AD. Twelve diseases associated with AD have been reported: high blood pressure, sleep apnea, obesity, hearing loss, depression and anxiety, vitamin deficiencies, down syndrome, type 2 diabetes, infections, gum disease, hypothyroidism, and environmental exposure.

**Figure 2 ijms-23-09703-f002:**
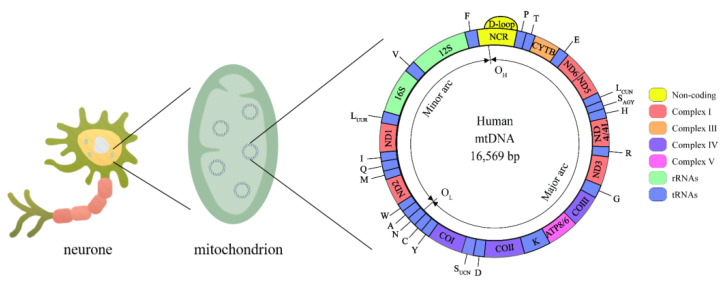
Mitochondrial genome. The human mtDNA genome is made up of 16,569 base pairs and consists of circular double-stranded DNA with an *L-strand* inside and an *H-strand* outside. mtDNA replication initiates within the *D-loop* region and proceeds from the *O_H_* until the *O_L_*. Complex I is encoded by *MT-ND1*, *MT-ND2*, *MT-ND3*, *MT-ND4*, *MT-ND4L*, *MT-ND5*, and *MT-ND6*, Complex III is encoded by *MT-CYTB*, complex IV is encoded by *MT-COI*, *MT-CO II*, and *MT-CO III*, complex V is encoded by *MT-ATP6* and *MT-ATP8*. rRNAs: *12S rRNA* and *16S rRNA*. tRNAs: *A*, *C*, *D*, *E*, *F*, *G*, *H*, *I*, *K*, *L_CUW_*, *L_UUR_*, *M*, *N*, *P*, *Q*, *R*, *S_AGY_*, *S_UCN_*, *T*, *V*, *W*, *Y*.

**Figure 3 ijms-23-09703-f003:**
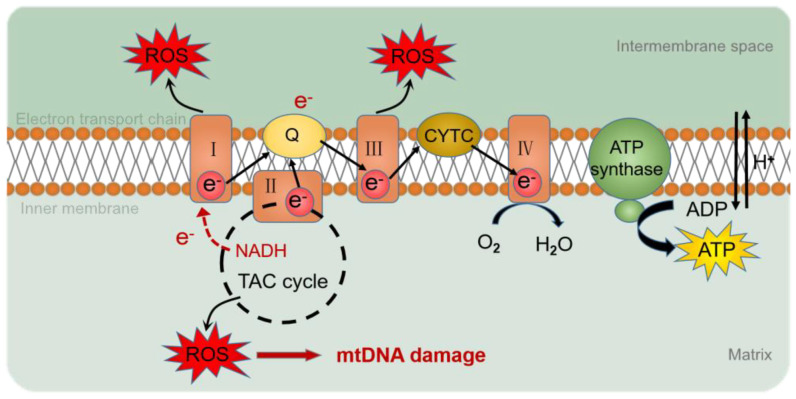
ROS production by the mitochondrial respiratory chain. During electron transfer in the mitochondrial respiratory chain, O_2_ or H_2_O_2_ is reduced to produce ROS. ROS accumulated in mitochondria cause oxidative damage to mtDNA. I: complex I; III: complex III; IV: complex IV; Q: ubiquinone; CYTC: cytochrome C; ROS: reactive oxygen species; e^-^: electron; NADH: nicotinamide adenine dinucleotide; TAC cycle: tricarboxylic acid cycle; O_2_: oxygen; H_2_O: hydrone; H^+^: hydrion.

**Figure 4 ijms-23-09703-f004:**
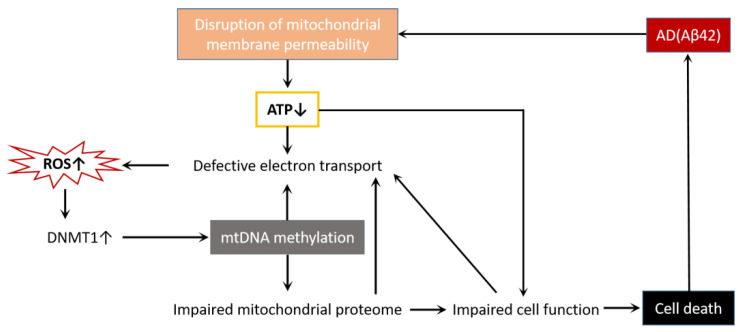
AD and cell death interact through mitochondrial dysfunction. Aβ42 disrupts mitochondrial membrane permeability, reduces ATP production, disrupts electron transport chains, and causes ROS accumulation. The increase in ROS activates DNMT1, leads to mtDNA methylation, impairs mitochondrial proteases, affects the normal function of cells, and finally causes cell death. Additionally, mtDNA methylation, mitochondrial protease impairment, and cell dysfunction all make the electron transport chain defective, forming a vicious cycle and exacerbating cell death.

**Figure 5 ijms-23-09703-f005:**
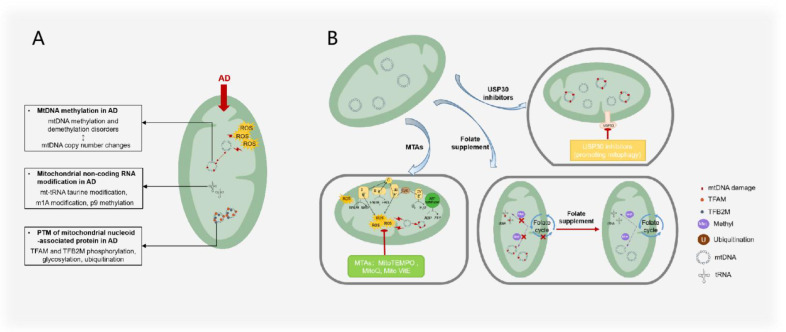
AD treatments and mechanisms that target mitochondrial epigenesis. (**A**) Mitochondrial epigenetic changes observed in AD mainly include the imbalance of mtDNA methylation and demethylation; tRNA taurine modification, m1A modification, and p9 methylation; phosphorylation, glycosylation, and ubiquitination of mitochondrial nucleoid proteins such as TFAM and TFB2M. mtDNA methylation and demethylation imbalance is correlated with mtDNA copy number changes. (**B**) AD treatment targeting mitochondrial epigenetics mainly includes the use of mitochondria-targeted antioxidants to reduce ROS damage to mtDNA, the use of folic acid supplements to maintain the methylation balance between mtDNA and RNA, and USP30 to inhibit the adverse ubiquitination of mitochondrial proteins.

**Table 1 ijms-23-09703-t001:** mtDNA methylation detection in different AD samples.

Sample	mtDNA Methylation	mtDNA Copy Number	Sequencing Methods	Observation	Reference
The hippocampi of APP/PS1 transgenic mice	The mtDNA methylations increase in *CYTB* and *COX II*	Decreases	Bisulfite pyrosequencing	Hypermethylation of the mitochondrial *CYTB* and *COX II* genes reduces mtDNA CN and gene expression in the APP/PS1 mouse model of AD.	[49]
The hippocampi of APP/PS1 transgenic mice	The mtDNA methylation reduces in the *D-loop* and increases in the *12S rRNA* gene	Decreases	Bisulfite pyrosequencing	Increased methylation levels of the *12S rRNA* gene decreased *12S rRNA* gene expression, decreasing mitochondrial biogenesis and mitochondrial dysfunction.	[59]
Entorhinal cortex in brain samples with Alzheimer’s disease-related pathology and cerebral cortex of APP/PS1 mice	The mtDNA methylation reduces in the *MT-ND1* gene and increases in the *D-loop*	/	454 GS FLX Titanium pyrosequencer	An increase in 5 mC in the promoter gene is usually accompanied by 5 mC damage to the coding region.	[60]
A previously described dataset of 263 subjects of Caucasian origin	The mtDNA methylation reduces in the *D-loop*	/	Methylation-sensitive high-resolution melting analysis	Concerning DNMT3A-448A>G polymorphisms, AA genotype carriers were only observed in patients with AD, and the level of *D-loop* methylation in AA genotype carriers was significantly higher than in GG and GA carriers.	[61]
Peripheral blood of LOAD samples	The mtDNA methylation reduces in the *D-loop*	/	Methylation-sensitive high-resolution melting analysis	The degree of methylation in the *D-loop* region may characterize different tissue or stages of disease in individuals with AD.	[62]
The peripheral blood cells of 14 mild cognitive impairment (MCI) patients, 18 early stage AD patients, 70 advanced-stage AD patients, and 105 healthy control subjects	*D-loop* methylation decreased in patients with AD in the late stages, while there was no significant change in the early stages, and *D-loop* methylation increased in patients with MCI in the late stages	/	Methylation-sensitive high-resolution melting analysis	*D-loop* methylation is inversely correlated with cerebrospinal fluid p-tau levels and age at the time of sampling. Mitochondrial methylation of the *D**-loop* is associated with different stages of AD, and these methylation changes are recognizable in the peripheral blood and therefore may provide biomarkers for the disease.	[63]
peripheral blood mononuclear cells	/	Decreases	/	CYT C, CYT B, PGC-1 α, and TFAM were decreased in AD	[64]

## Data Availability

Not applicable.

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
