# Peer review of "Targeted Mitochondrial Epigenetics: A New Direction in Alzheimer’s Disease Treatment"

_ijms, 2022, doi:10.3390/ijms23179703_

Round 1
Reviewer 1 Report
This article summarized epigenetics of mitochondrial DNA and the role this might play in AD. There is also a review of potential therapeutics which target epigenetic changes in DNA. There are some concerns regarding missing information in the review article. There is a table describing prior studies which examined mtDNA methylation at specific DNA sequences, but its unclear of the rigor of this research. One difficulty in this field has been and will continue to be mitochondrial DNA pseudogenes present in the nucleus. How did these cited studies overcome that? How do they know for certain they are detecting true authentic mitochondrial DNA and not nuclear psuedogenes? I think these caveats need to be addressed in this review.
It would also be helpful to the readers for the review article to be edited for style and language.
Author Response
The studies presented in Table 1 were sequenced using bisulfite pyrosequencing, 454 GS FLX Titanium pyrosequencing, methylation-sensitive high-resolution melting (MS-HRM) methods, respectively. The 454 GS FLX Titanium pyrosequencing process assesses methylation at CpG and non-CpG sites. The number of unmethylated reads was higher than the number of methylated reads per identified site, and few methylated sites were lost. Therefore, the researchers removed from the analysis those reads that showed at least one missing site in the methylation pattern after alignment to avoid bias in quantification. This method effectively avoids the analysis of putative mitochondrial pseudogenes and ensures that the amplicons are 100% identical to mtDNA. bisulfite pyrosequencing and MS-HRM also need to eliminate the influence of mitochondrial pseudogenes by certain means , such as isolating mitochondria before mtDNA extraction to avoid nuclear contamination, and specific primers considering NUMT amplification Basic Local Alignment Search Tool searches to identify known NUMTs.
Reviewer 2 Report
In this manuscript, the authors summarized the current understanding of the mechanisms of mitochondrial epigenetic alterations contributing to Alzheimer's disease; discussed the pathological changes in neurodegenerative disease associated with mitochondrial epigenetic alterations; outlined possible pathways of mitochondrial epigenetic changes in AD; also discussed the further treatment of AD by mediating mitochondrial epigenetics. This study is overall well-designed, well written, and the literatures are well interpreted. Few minor issues need to address:
- The massive literatures were well interpreted in the main text, however, the citation source is supposed to be cited accurately. The authors may consider going over all the references to make sure those all are cited properly. For example, reference 8 (PMID: 33188775) investigated Tau isoforms and post-translational modification stoichiometry instead of mitochondrial epigenetic in Alzheimer's disease , even if there is high relevance between mitochondria epigenetic and Tau protein for AD development. There is no need
- The figures are extraordinary, however, a detailed legend instead of a single title with few words would be helpful to better understand the contents.
- For the section of Conclusions and Future Perspectives, the authors discussed the advantage of mitochondrial epigenetic treatment for AD, however, lacking discussion about the limitations of current investigations and disadvantages of treatment with mitochondrial-targeted approaches weakens the future perspectives.
Author Response
1. I removed references that were less relevant to the content of the article, 8,57,58,59,62,64, and duplicate references 70,118.
2. Captions have been added to each figure.
Figure 1. Related diseases that increase the risk of developing AD. Twelve diseases associated with AD have been reported: high blood pressure, sleep apnea, obesity, hearing loss, depression and anxiety, vitamin deficiencies, down syndrome, type 2 diabetes, infections, gum disease, hypothyroidism and environmental exposure.
Figure 2. Mitochondrial genome. The human mtDNA genome is made up of 16,569 base pairs and consists of circular double-stranded DNA with an L-strand inside and an H-strand outside. MtDNA replication initiates within the D-loop region and proceeds from the OH until the OL.
Figure 3. ROS production by the mitochondrial respiratory chain. During electron transfer in the mitochondrial respiratory chain, O2 or H2O2 is reduced to produce ROS. ROS accumulated in mitochondria cause oxidative damage to mtDNA.
Figure 4. AD and cell death interact through mitochondrial dysfunction. Aβ42 disrupts mitochondrial membrane permeability, reduces ATP production, disrupts electron transport chains, and causes ROS accumulation. The increase in ROS activates DNMT1, leads to mtDNA methylation, impairs mitochondrial proteases, affects the normal function of cells, and finally causes cell death. Additionally, mtDNA methylation, mitochondrial protease impairment and cell dysfunction all make the the electron transport chain defective, forming a vicious cycle and exacerbating cell death.
Figure 5. AD treatments and mechanisms that target mitochondrial epigenesis. A) Mitochondrial epigenetic changes observed in AD mainly include the imbalance of mtDNA methylation and demethylation; tRNA taurine modification, m1A modification and p9 methylation; phosphorylation, glycosylation and ubiquitination of mitochondrial nucleoid proteins such as TFAM and TFB2M. MtDNA methylation and demethylation imbalance is correlated with mtDNA copy number changes. B) AD treatment targeting mitochondrial epigenetics mainly includes the use of mitochondria-targeted antioxidants to reduce ROS damage to mtDNA, the use of folic acid supplements to maintain the methylation balance between mtDNA and RNA, and USP30 to inhibit the adverse ubiquitination of mitochondrial proteins.
3. The limitations of current investigations and disadvantages of treatment with mitochondrial-targeted approaches are added in the section of Conclusions and Future Perspectives.
"However, there are still some difficulties that cannot be ignored in mitochondrial epigenetic studies, such as false-positive and false-negative results of mtDNA methylation caused by the mtDNA secondary structure and mitochondrial pseudogenes [112]. How to be sure that what is being detected is true mtDNA, rather than mitochondrial pseudogenes, is a challenge researchers must overcome. Numts contamination is an important reason for the controversial application of bisulfite-pyrosequencing to mtDNA [113]. The detection of mtDNA methylation by bisulfite-pyrosequencing requires certain means to eliminate sequences with uncertain reads due to numts. Next-generation sequencing technology, such as 454 GS FLX titanium pyrosequencing, can effectively eliminate the influence of mitochondrial pseudogenes on the detection results, but this requires better equipment. Therefore, the development of more convenient, inexpensive and efficient mtDNA methylation detection methods is the key to promoting the progress of mitochondrial epigenetics in AD."
"
However, the above treatment routes have shortcomings: (1) The duration of folic acid supplementation treatment is short, the optimal folic acid dose is unclear, and folic acid may need to be synergistic with other AD treatment drugs[114]; (2) although USP30 inhibitors have been shown to alleviate mitophagy impairment in Parkinson’s syndrome [115], it remains to be demonstrated whether USP30 inhibitors are effective in AD."
.

Round 2
Reviewer 1 Report
concerns have been addressed